# The Mental Health Burden of Patients with Colorectal Cancer Receiving Care during the COVID-19 Pandemic: Results of the PICO-SM Study

**DOI:** 10.3390/cancers15041226

**Published:** 2023-02-15

**Authors:** Kok Haw Jonathan Lim, Panagiotis Ntellas, Daniel Anderson, Lilly Simpson, Michael Braun, Marios Adamou, Jorge Barriuso, Katerina Dadouli, Jacqueline Connell, Joseph Williams, Theodora Germetaki, Deirdre Lehwald, Niall Fitzpatrick, Mark Cutting, Danielle McCool, Jurjees Hasan, Saifee Mullamitha, Kalena Marti, Mark Saunders, Konstantinos Kamposioras

**Affiliations:** 1Department of Medical Oncology, The Christie NHS Foundation Trust, Manchester M20 4BX, UK; 2Department of Medical Oncology, University Hospital of Ioannina, 45500 Ioannina, Greece; 3Department of Psycho-Oncology, The Christie NHS Foundation Trust, Manchester M20 4BX, UK; 4School of Human and Health Sciences, Queensgate, University of Huddersfield, Huddersfield HD1 3DH, UK; 5Division of Cancer Sciences, Faculty of Biology, Medicine and Health, University of Manchester, Manchester M13 9NT, UK; 6Manchester Cancer Research Centre, Manchester M20 0GJ, UK; 7Laboratory of Hygiene and Epidemiology, Faculty of Medicine, University of Thessaly, 41222 Larissa, Greece; 8Division of Pharmacy and Optometry, School of Health Sciences, University of Manchester, Manchester M13 9PT, UK; 9Department of Clinical Oncology, The Christie NHS Foundation Trust, Manchester M20 4BX, UK

**Keywords:** COVID-19, anxiety, depression, well-being, psycho-oncology

## Abstract

**Simple Summary:**

The COVID-19 pandemic has resulted in unprecedented changes to the life of patients with cancer. In this study, we aim to evaluate the impact of the COVID-19 pandemic on the mental health and general well-being of patients with colorectal cancer by carrying out a prospective longitudinal questionnaire. We found that around one in four participants reported symptoms of anxiety and poor well-being, with 15% at risk of moderate to severe depression. Amongst others, those who were worried that the COVID-19 pandemic would have an effect on their mental health were most at risk of anxiety, depression, and poor well-being. Screening for the mental health impact of the COVID-19 pandemic on patients is essential to allow timely action from all key stakeholders in order to avoid potentially longer-term detrimental consequences.

**Abstract:**

The COVID-19 pandemic has resulted in unprecedented changes to the lives of patients with cancer. To evaluate the impact of the COVID-19 pandemic on the mental health and well-being of patients with colorectal cancer, we conducted a prospective longitudinal questionnaire study at a UK tertiary cancer centre. In total, 216 participants were included: mean age 65 years, 57% (*n* = 122) male, 92% (*n* = 198) of white ethnicity. Amongst participants who completed the screening psychometric questionnaire, 24% (*n* = 48/203) reported anxiety (GAD-7 ≥ 5), 15% (*n* = 31/204) depressive symptoms (PHQ-9 ≥ 10), 3% (*n* = 5/190) probable post-traumatic stress disorder (PC-PTSD-5 ≥ 4), and 31% (*n* = 66/213) poor well-being (WHO-5 < 50). In the subgroup (*n* = 95/216, 44%) who consented to and completed a follow-up survey 6 months later, there was a significant increase in the number of participants at risk of depression (4% vs. 13%, *p* = 0.021). Self-reported concern about the COVID-19 pandemic impacting one’s mental health is associated with increased likelihood of anxiety, depression, and poor well-being, in respective multivariate analyses. In conclusion, screening for the mental health impact of the COVID-19 pandemic is essential to ensure timely action from all key stakeholders and to avoid potentially longer-term detrimental consequences.

## 1. Introduction

The coronavirus disease 2019 (COVID-19) pandemic has resulted in significant changes in the management of patients diagnosed with cancer [1,2]. Several mitigating measures to hospital visits and treatment modifications were initially rapidly enforced, in part to minimise risks of nosocomial infection and to prioritise safe clinical management of patients most in need. During the first phase of the pandemic in early 2020, we conducted a survey study at The Christie NHS Foundation Trust (Manchester, UK), a tertiary cancer centre in North West England, to assess how changes in service delivery had been perceived by patients with colorectal cancer [3,4]. This highlighted that at least one in five patients were at risk of anxiety, and amongst others, the key priority for most patients was to carry on with their cancer treatments [3]. However, in these earlier studies, the focus was to identify areas for service improvement to better support patients during the initial crisis period, and therefore detailed insights into the impact of the COVID-19 pandemic on the mental health of patients was limited.

Despite significant progress brought about by national vaccination campaigns, most countries worldwide continue to be affected by the emergence of new strains of the severe acute respiratory syndrome coronavirus 2 (SARS-CoV-2), resulting in significant pressures on the healthcare systems [5]. Some vulnerable patient groups continue to experience ongoing restrictions to their daily lives and cancer care journey [6,7], which would inevitably have immediate and longer-term consequences on their mental health. The ‘Psychological Impact of COVID-19 on Patients with Solid Malignancies: A Single-Institution Survey Study’ (PICO-SM) is a prospective longitudinal study designed to investigate the mental health burden of patients with cancer, assessing the prevalence and risk of anxiety, depression, post-traumatic stress disorder (PTSD), as well as general well-being. Here, we report the results of PICO-SM study, highlighting the psychological burden of the COVID-19 pandemic specifically in patients with colorectal cancer.

## 2. Materials and Methods

### 2.1. Participants

PICO-SM is a single centre prospective survey study conducted at a large comprehensive cancer centre in North West England, UK (The Christie NHS Foundation Trust). All participants were aged ≥ 18 years with a diagnosis of colorectal cancer and who were able to fully comprehend the patient information sheet were invited to participate. Participants who lacked capacity were excluded. Participation was entirely voluntary, and no financial incentive was offered for completion of the survey. 

### 2.2. Survey Design

The survey is a 30-item questionnaire, including questions on basic demographics, the current status of the individual participant’s cancer, participants’ perception of their treatment, risk factors for COVID-19 infection, and the impact of the COVID-19 pandemic on their mental health. Participants were also questioned regarding coping strategies used (if any) during the pandemic. This survey was designed by a multidisciplinary group of experienced oncologists, psychologists, psychiatrists, nurse specialists, and with input from patient representatives.

### 2.3. Study Measures

The primary objective of PICO-SM was to evaluate the levels of anxiety, depression, post-traumatic stress (PTSD), and general well-being amongst patients with colorectal cancer during the COVID-19 pandemic. The following validated self-reported screening tools for the presence and/or severity of each of the outcomes of interest were used: Generalized Anxiety Disorder scale (GAD-7) for anxiety [8], Patient Health Questionnaire-9 (PHQ-9) for depression [8,9], Primary Care Post-Traumatic Stress Disorder-5 (PC-PTSD-5) for probable PTSD [10], and World Health Organization Well-being Index (WHO-5) for mental well-being [11,12]. Our secondary objective was to understand the wider needs of patients with cancer during the period of a pandemic in order to help further develop immediate and/or longer-term support measures.

### 2.4. Implementation

Potential participants were identified through clinic list review (specialized colorectal oncology service) and were recruited in person or remotely (via telephone or video consultations) between 7 and 28 April 2021 (timepoint 1). Printed copies of the approved versions of the participant information sheet and survey were used. Participants had the option of either completing the survey onsite or return responses using a prepaid envelope by post. Consent was obtained upon return of the completed survey. Participants were asked to indicate if they agree to be contacted again on a future date, and those who consented to this were subsequently approached again 6 months after the day of the completion of the initial survey, between 7 October 2021 and 21 January 2022 (timepoint 2). Participants were also offered the option of being contacted for a Psycho-Oncology referral for additional support should their responses score above a certain threshold.

### 2.5. Statistical Analysis

Descriptive analysis for all the variables was carried out using IBM SPSS Statistics for Windows, version 26 (IBM Corp, Armonk, NY, USA) and data represented using Microsoft Excel and GraphPad Prism version 9.3.0 for Mac (San Diego, CA, USA). Continuous variables were expressed as means ± standard deviations, and categorical variables were expressed as frequencies and percentages. Categorical data were analysed with the use of Chi-square test or Fisher’s exact test. Continuous variables were checked for deviation from normal distribution (Kolmogorov–Smirnoff normality test) for each comparison. Student’s *t*-Test or Mann–Whitney U Test were performed for continuous data as appropriate. To compare the participants’ answers between timepoints 1 and 2 of the longitudinal survey study, we conducted McNemar test or related-samples Wilcoxon signed-rank test.

Multivariate analysis was performed using a logistic regression for binary outcomes (anxiety (GAD-7 score ≥ 5), depression (PHQ-9 score ≥ 10), and poor well-being (WHO-5 < 50)), and odds ratios (OR) with 95% confidence intervals (CIs) were calculated. Variables with *p* < 0.10 on univariate analyses were included in the final model, respectively. Multivariable analysis was not conducted for PTSD due to low incidence of events. Pearson bivariate correlation analysis was used to validate the association between the key outcome measures used: GAD-7, PHQ-9, PC-PTSD-5, and WHO-5. For all the analyses, a 5% significance level was set, and *p* values were two-tailed.

## 3. Results

### 3.1. Survey Participants

A total of 473 consecutive patients with colorectal cancer were identified and invited to participate in the PICO-SM study. At the end of the study enrolment window, 216 patients (response rate 45.7%) were included in the final analysis of the primary cohort (timepoint 1). One hundred and sixty-five patients were invited to participate in the longitudinal follow-up survey 6 months later, as 51 patients were either too clinically unwell to be contacted again or had died during the follow-up period of the study. A total of 95 patients (57.5%) subsequently consented to and completed the survey at timepoint 2. 

The mean age of the index cohort was 65 years and included 122 (56.5%) male participants and 198 (91.7%) of white ethnicity (Table 1). One in five (*n* = 42/216, 19.4%) participants disclosed that they have an underlying mental health condition, with anxiety (*n* = 25/216, 11.6%) and depression (*n* = 18/216, 8.3%) being the most prevalent (Table 1). Importantly, 31.0% (*n* = 67/216) of participants identified themselves as having a condition and/or comorbidity, which may put them at higher risk if they acquire COVID-19 infection (e.g., diabetes, respiratory, or circulatory morbidity) (Appendix A). At the time of the survey, the majority of participants (*n* = 171/216, 79.2%) had had a test for COVID-19, with 9 (5.3%) participants disclosing that they had had a positive test (Appendix A). Three patients had required hospitalisation for COVID-19 (Appendix A). 

### 3.2. The Mental Health Impact of the COVID-19 Pandemic

The majority (*n* = 155/216, 71.8%) of the participants felt that the COVID-19 pandemic did not affect their mental health at all or only slightly (Table 2). Amongst participants who completed the screening psychometric questionnaire, 23.6% (*n* = 48/203) were at risk of mild to severe anxiety (GAD-7 score ≥ 5), 15.2% (*n* = 31/204) reported moderate to severe depressive symptoms (PHQ-9 score ≥ 10), 2.6% (*n* = 5/190) had probable PTSD (PC-PTSD-5 score 4–5), and 31.0% (*n* = 66/213) had poor well-being (WHO-5 < 50) (Figure 1). Additionally, we conducted an internal validation testing and confirmed a statistically significant (*p* < 0.05) correlation between the various psychometric screening tools used in this study. There was a strong correlation between GAD-7 and PHQ-9 (*r* = 0.680, *p* < 0.001) and WHO-5 and GAD-7 (*r* = −0.558, *p* < 0.001) or PHQ-9 (*r* = −0.746, *p* < 0.001). There was a weak to moderate correlation between PC-PTSD-5 and WHO-5 (*r* = −0.193, *p* = 0.008), GAD-7 (*r* = 0.328, *p* < 0.001), or PHQ-9 (*r* = 0.286, *p* < 0.001).

During the initial period of this survey study, more than half (*n* = 132/216, 61.1%) of participants were not at all or only slightly concerned about contracting COVID-19, and only a minority (*n* = 58/216, 26.9%) felt that the COVID-19 pandemic had/will have a negative impact on their cancer treatment (Appendix A). The large majority (*n* = 193/216, 89.4%) were more worried about their cancer rather than the risk of COVID-19 infection (Appendix A). The top three main concerns for most were disease relapse or progression while waiting for treatment (*n* = 37/216, 17.1%), knowing where to get help with dealing with side effect (*n* = 16/216, 7.4%), and uncertainty around when treatment or tests will restart (*n* = 14/216, 6.5%) (Appendix A). 

### 3.3. Subgroup Analysis: Longitudinal Changes in the Mental Health Impact of the COVID-19 Pandemic

We tracked the responses to the screening psychometric tools for the subgroup of *n* = 95 patients who completed a further follow-up survey (same questionnaire) at timepoint 2. Within this longitudinal cohort, there were no differences in the proportion of participants who were at risk of poor well-being (25.3% vs. 22.1%), anxiety (17.9% vs. 20.0%), or PTSD (4.3% vs. 4.3%) (Figure 2). However, over time, there was a statistically significant increase in the number of participants who scored 10 or more at the PHQ-9 for depression (4.3% vs. 12.6%, *p* = 0.021) (Figure 2). Paired comparison across all the characteristics of participants between the two timepoints revealed that the only difference was a doubling in the number of participants who in the interim had developed progressive disease (*n* = 7, 7.4% vs. *n* = 18, 18.9%; Related-samples Wilcoxon signed-rank test, *p* = 0.015) (Appendix A). Moreover, there was a statistically significant decline in the proportion of participants who felt supported by their GP (*n* = 49, 51.6% vs. *n* = 43, 45.3%; Related-samples Wilcoxon signed-rank test, *p* = 0.004) or the government (*n* = 50, 52.6% vs. *n* = 39, 41.1%; Related-samples Wilcoxon signed-rank test, *p* = 0.010) (Appendix A).

### 3.4. Coping Strategies and Support Received during the COVID-19 Pandemic

Various self-coping strategies were reportedly being used by the participants during the COVID-19 pandemic. Focussing on positives (*n* = 115/216, 53.2%), the use of humour (*n* = 84/216, 38.9%), change in physical activity (e.g., exercise) (*n* = 73/216, 33.8%), and avoiding thinking about the pandemic (*n* = 66/216, 30.6%) were most commonly reported (Table 2). Almost all (*n* = 200/207; 97.1%) patients reported that they were well-supported by their families and/or friends (Figure 3). In addition, participants also reported that they were moderately to extremely well-supported by their care providers as follows: specialist cancer team (*n* = 191/204, 93.6%), nurse specialists (*n* = 141/175, 80.6%), general practitioners (GP) (*n* = 114/192, 59.4%), and community services (*n* = 96/174, 55.2%) (Figure 3). Just over half (*n* = 97/187, 51.9%) of the participants felt moderately to extremely well-supported by the government (Figure 3). Overall, the majority of the participants did not feel they needed further support for their mental health (*n* = 192/216, 88.9%) (Table 2).

### 3.5. Factors Associated with Anxiety, Depression, and Poor Well-Being

In the multivariate analysis model for the whole primary cohort studied, anxiety (GAD-7 score ≥ 5) was associated with those who were worried that the COVID-19 pandemic would have an effect on their mental health (OR 2.26, 95% CI: 1.32–3.88, *p* = 0.003), those whose self-reported disease status is unknown/uncertain (OR 5.16, 95% CI: 1.56-17.10, *p* = 0.007), and those who wanted more support (OR 9.77, 95% CI: 1.39–68.93, *p* = 0.022) (Table 3). Meanwhile, those who self-declared as not having any past history of mental health condition were less likely to have anxiety (OR 0.17, 95% CI: 0.06–0.50, *p* = 0.001) (Table 3). 

Similarly, those who were worried that the COVID-19 pandemic would have an effect on their mental health were also more likely to have moderate to severe depressive symptoms (PHQ-9 score ≥ 10) (OR 6.18, 95% CI: 1.15–33.03, *p* = 0.033), as did using ‘distracting self’ as a coping strategy (OR 5.57, 95% CI: 1.12–27.82, *p* = 0.036) (Table 3). On the other hand, white ethnicity (OR 0.12, 95% CI: 0.02–0.90, *p* = 0.039) and ‘focusing on positives’ as a coping strategy (OR 0.04, 95% CI: 0.01–0.25, *p* < 0.001) were both negatively associated with the risk of moderate to severe depression (Table 3). 

Factors which were associated with poor well-being (WHO-5 < 50) on multivariate analysis included those who were worried that the COVID-19 pandemic would have an effect on their mental health (OR 2.44, 95% CI: 1.58–3.76, *p* < 0.001), and ‘change in substance intake’ as a coping mechanism (OR 7.69, 95% CI: 1.25–47.19, *p* = 0.028) (Table 3). ‘Using humour’ as a coping strategy (OR 0.02, 95% CI: 0.09–0.61, *p* = 0.003) and support from friends and/or family (OR 0.52, 95% CI: 0.33–0.83, *p* = 0.006) were inversely associated to poor well-being (Table 3).

In an exploratory multivariate analysis modelled based on the responses from the subgroup of participants who completed the follow-up study at timepoint 2, there were several other different or additional factors which were associated with risks of anxiety, depression, and poor well-being (Appendix A). Critically, at timepoint 2, concerns about contracting COVID-19 appeared to be associated to anxiety (OR 4.42, 95% CI: 1.23–15.88, *p* = 0.023), depression (OR 18.17, 95% CI: 2.33–141.92, *p* = 0.006), and poor well-being (OR 4.18, 95% CI: 1.71–10.17, *p* = 0.002) (Appendix A).

## 4. Discussion

The delivery of cancer care continues to be affected by the unrelenting surges and waves of the COVID-19 pandemic more than two years on since the declaration of a global pandemic by the World Health Organization (WHO) [5]. Numerous comprehensive guidelines have been produced by national and international oncological societies to ensure the continuity of oncological management whilst maintaining patient safety [13]. However, there is now a palpable sense of fatigue amongst the general population, not least amongst vulnerable people, including patients with cancer. Indeed, there is also an increasing number of reports on the urgent need for psychological support for both patients and healthcare workers during this acute period of crisis but also in the longer term [14,15].

The PICO-SM study confirms some of the key findings of our previous survey study conducted during the first phase of the COVID-19 pandemic [3]. Cancer care remains the key priority for patients, and most did not appear to be significantly distressed by the pandemic. Continuation of systemic treatment and cancer-related distress were also reported to be the main concerns for patients with cancer in several other studies [16,17,18]. In both survey studies conducted at our institute [3], the majority of participants (89%) did not feel that they needed support for matters directly related to COVID-19; their main concern was that their cancer did not recur or progress while awaiting for treatment to start. By the time of this present study, it appears that patients had already become accustomed to the nuances of the pandemic and the rationale of some of the necessary changes in the delivery of cancer care, and thus did not perceive that their cancer treatment would be affected by the pandemic (60%), in contrast to the initial survey where there was a high proportion (72%) who felt uncertain how their treatment could be affected [3].

More than one in five participants had reported symptoms of anxiety (GAD-7 score ≥ 5), similar to 2020 levels; however, in this more recent survey, the levels of moderate to severe anxiety levels had doubled (11.3% vs. 5.5% in 2020) [3]. In our initial report, we proposed a simple methodology to screen for patients who may be at higher risk of anxiety [3]. We identified that those who had concerns about contracting the COVID-19 infection, and worried that the pandemic would have an effect on their mental health and affect the experience of cancer care for those most at risk of anxiety [3]. In the PICO-SM study, we found that those who expressed concerns that the COVID-19 pandemic would have effect on mental health continue to be at highest risk for anxiety. While the fear of acquiring COVID-19 infection appeared to have been alleviated initially, this prevailed again as it was associated with an increased risk of anxiety, depression, and poor well-being in the subgroup of patients who were followed-up 6 months after the primary study period. Collectively, this further shows that patients who express concerns about the impact of the COVID-19 pandemic on their mental health, especially those with underlying mental health conditions, should be offered further screening and tailored support to ensure that their mental health is best catered for. 

Further, the PICO-SM study also interrogated the wider impact of the COVID-19 pandemic on other mental health aspects of patients with colorectal cancer. We identified that 24% of participants reported symptoms of anxiety, with 15% at risk of moderate to severe depression. This is similar to levels described in a recent meta-analysis of anxiety and depression among patients with cancer [19], although the majority of the studies that used PHQ-9 as the screening tool included Asian populations, and patients with colorectal cancer were underrepresented. The prevalence of clinically significant depression in patients with surgically treated early-stage colorectal cancer has been reported to vary between 21% preoperatively to 14% five years after surgery, indicating the need for long term follow-up even in patient with curative disease [20]. In our main multivariate analysis model, specific coping strategies used by participants were associated with depression. Interestingly, those who used self-distraction as a coping mechanism were most at risk of depression, while those focusing on positives had a lower risk of depression.

We also assessed patients’ general well-being using the validated WHO-5 Well-Being Index and found that 31% had poor well-being. This is similar to levels observed in the general population in a recent nationwide cross-sectional convenience-sampling questionnaire study conducted in Austria [21]. In our study, participants who were worried that the COVID-19 pandemic would have effect on mental health and change in substance intake were at higher risk of poor well-being. Since our initial study in 2020 [3], we recognised that the lack of contact with clinical team is a stressor for anxiety, and this might be related to a perception of loneliness and isolation [22]. Therefore, to alleviate patients’ concerns regarding the COVID-19 pandemic, including on their mental health, our centre provided a 24/7 COVID-19-specific ‘Hotline’ for patients to call if they have any concerns, in addition to directly contacting their named cancer-specific clinical nurse specialists.

The prevalence of PTSD was low (3%) among the participants in the PICO-SM study, which is significantly lower than the 21% reported in a prospective French study by Joly et al. [16]. In contrast to that COVIPACT study [16], where the prevalence of PTSD was assessed during the initial lockdown in France, in our study, PTSD was assess a year later when the national restrictive measures were already relaxed in the UK. Moreover, they used a different screening tool for PTSD and studied a more diverse population, with various cancer subtypes included [16]. Notably, patients with gastrointestinal malignancies had among the lowest proportion of PTSD (14%), although still higher compared to our study. 

Overall, there was no specific demographic characteristics found to be related with higher levels of distress. Ethnic minorities, younger age, and female gender have been previously reported to be factors related to psychological distress [23]. Ethnic minorities were underrepresented in our participant cohort to be able draw any definitive conclusions, and the latter two covariates were not associated with higher distress in our cohort. Pre-existing mental health conditions have been previously correlated with worse oncological outcomes, as clinicians were thought to be more reluctant to offer aggressive treatments and patients may report cancer symptoms at a more advanced stage [24]. In our study, about one in five participants self-reported as having an underlying mental health condition, with anxiety being the most common. Although we were not able to confirm that these patients indeed had a formal diagnosis of a mental health condition, it appears that, collectively, self-identification of the respective conditions may constitute a risk factor for anxiety and depression.

Strong social support has been generally linked with increased psychological resilience after a global disaster [25]. In our cohort, all but a few participants felt moderately to extremely well-supported by their friends and family, which might account for why the majority did not feel they needed more support. However, healthcare authorities should not rest on their laurels and continue to maintain high levels of planning and preparedness to meet patients’ demands during any potential crisis period in the future. Finally, several other coping mechanisms have been reported in the literature to mitigate the impact of the pandemic for both patients and health care professionals [15,22,26,27]. Humour and sarcasm are recognised ways to cope with stressful conditions, although humour use might be used more frequently in the presence of an underlying psychiatric condition [28]. Positive thinking, meditation, and increased exercise are also other common methods associated with stress relief among non-cancer patients [22]. Clinical teams should be aware of the repertoire of such commonly used and effective coping mechanisms and be equipped to offer personalised and open discussions on these to maximise the holistic care for their patients, as is appropriate for their respective circumstances.

Despite the strengths of our prospective study, one must acknowledge the inherent limitations of a self-reported survey study. This study was conducted at a tertiary cancer centre, which may not reflect the practices in smaller centres where support services might be more limited (e.g., lack of psycho-oncology services, no dedicated clinical nurse specialists, etc.). The response rate to this survey study (46%) was lower than expected, which may be reflective of the stresses and other priorities for patients during the COVID-19 pandemic and may therefore result in an underestimation of the overall scale of the mental health impact described here. As discussed previously, we have also noted an underrepresentation of ethnic minorities engaged in our survey. This is, however, representative of the demographic composition of the North West of England’s general population (catchment area of our Colorectal Oncology Service), of whom 86% are of ‘White’ ethnicity (Census 2021, Office for National Statistics). In addition, the PICO-SM study was only open for recruitment for three weeks, and this might have further compounded on the unintentional stochastic sampling bias. Therefore, the potential mental health issues or coping strategies used by this important subgroup of patients remain poorly understood by our study. We inferred some longitudinal comparisons to a previous study conducted at our centre [3], and although the participants recruited in the PICO-SM study were different to those in our index study, there was consistency in the responses to some of the key questions, indicating that the patient population receiving their cancer care under the same service may have had similar experience and support throughout the pandemic. Subsequent to this, as pre-planned within the PICO-SM study protocol, we will undertake further follow-up of a subgroup of participants whose responses were analysed in this study.

## 5. Conclusions

In conclusion, the results from the PICO-SM study show that the COVID-19 pandemic has had a significant impact on the mental health burden of patients with colorectal cancer. Corroborating with our previous work, participants who reported concerns about the COVID-19 pandemic impacting on their mental health are at greatest risk, and therefore, this may be a practical screening tool which may be easily deployed in routine clinical practice. The real significant longer-term and wider impact of the pandemic on patients receiving care during these difficult times and their families remain unknown. The implications of our findings will need to be validated in a larger cohort of patients, including in patients with other types of malignancy, and in other severe or chronic diseases. This is vital in appropriately planning and costing health care delivery to ensure that robust support services are instituted as nations continue their efforts in making recovery plans going forward. 

## Figures and Tables

**Figure 1 cancers-15-01226-f001:**
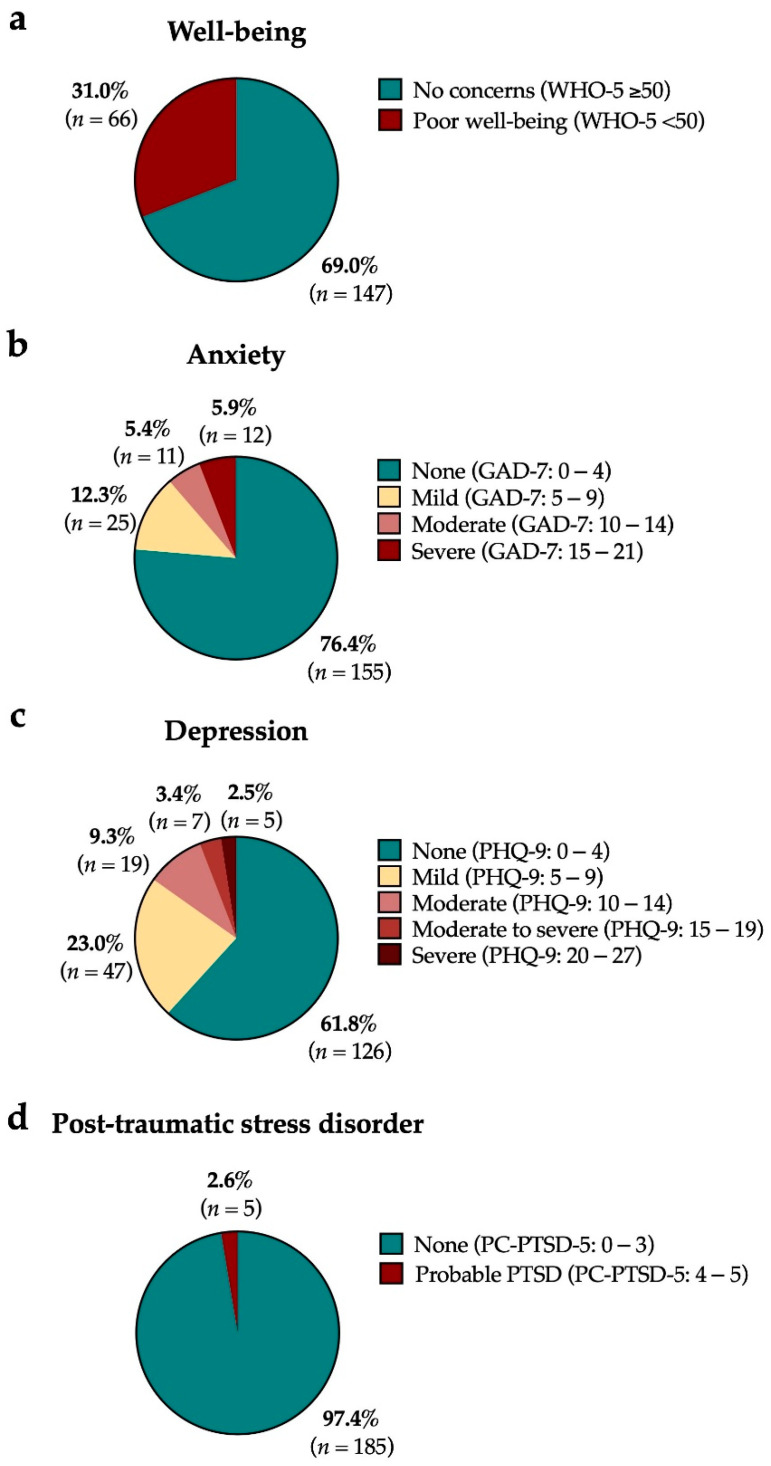
The impact of the COVID-19 pandemic on the levels of (**a**) poor well-being, (**b**) anxiety, (**c**) depression, and (**d**) post-traumatic stress disorder (PTSD) in patients with colorectal cancer (*n* = 216). Abbreviations: GAD-7, Generalized Anxiety Disorder scale; PC-PTSD-5, Primary Care Post-Traumatic Stress Disorder-5; PHQ-9, Patient Health Questionnaire-9; WHO-5, World Health Organization Well-being Index.

**Figure 2 cancers-15-01226-f002:**
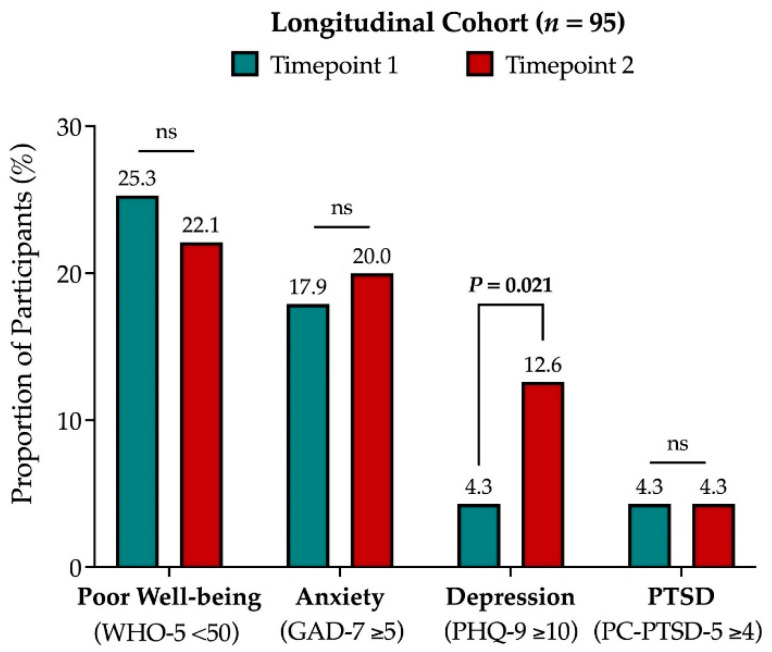
Comparison of the key outcome variables of the PICO-SM study (anxiety, depression, PTSD, and poor wellbeing) for the longitudinal subgroup of *n* = 95 participants followed-up across two timepoints, 6 months apart. Abbreviation: PTSD, post-traumatic stress disorder; ns, not significant.

**Figure 3 cancers-15-01226-f003:**
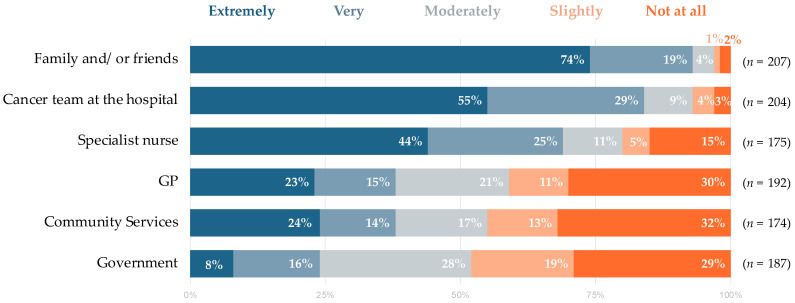
Perception of levels of support participants felt they had received during the COVID-19 pandemic.

**Table 1 cancers-15-01226-t001:** Baseline demographics of participants in the PICO-SM study (*n* = 216).

	Number, *n* (%)
**Gender**	
Male	122 (56.5)
Female	91 (42.1)
Other	1 (0.5)
*Prefer not to say*	2 (0.9)
**Mean age (years)**	65.3 ± 10.2 years
**Ethnicity**	
White/White British	198 (91.7)
Asian/Asian British (Indian, Pakistani, Bangladeshi)	5 (2.3)
Asian/Asian British (Chinese)	1 (0.5)
Black/Black British	3 (1.4)
Mixed	2 (0.9)
Other	5 (2.3)
*Prefer not to say*	2 (0.9)
**Marital Status**	
Single/ Divorced /Separated/ Widowed	70 (32.4)
In a relationship/Married/In civil partnership	144 (66.7)
*Prefer not to say*	2 (0.9)
**Have children**	
Yes	168 (77.8)
No	44 (20.4)
*Prefer not to say*	4 (1.9)
**Lives alone**	
Yes	51 (23.6)
No	162 (75.0)
*Prefer not to say*	3 (1.4)
**Previous/Underlying diagnosis of mental health condition ^a^**	
Yes	42 (19.4)
Anxiety	25 (11.6)
Depression	18 (8.3)
Panic attacks	9 (4.2)
Anorexia	0 (0.0)
Psychosis	0 (0.0)
Bulimia	0 (0.0)
Social phobia	1 (0.5)
Attention Deficit disorder	1 (0.5)
Obsessive compulsive disorder	0 (0.0)
Autism	0 (0.0)
Post-traumatic stress disorder	2 (0.9)
Alcohol/drugs	1 (0.5)
Bipolar disorder	1 (0.5)
Personality disorder	1 (0.5)
Other	1 (0.5)
None of the above	169 (78.2)
**Self-reported perception of current status of cancer**	
Stable disease	126 (58.3)
Progressive disease	33 (15.3)
Unknown	57 (26.4)

^a^ more than one option to question possible.

**Table 2 cancers-15-01226-t002:** The general impact of the COVID-19 pandemic on patients with colorectal cancer and support/coping strategies used (*n* = 216).

	Number, *n* (%)
**Felt COVID-19 pandemic has affected mental health**	
Extremely	3 (1.4)
Very much	11 (5.1)
Moderately	39 (18.1)
Slightly	53 (24.5)
Not at all	102 (47.2)
*Did not answer*	8 (3.7)
**Mental health has affected experience of cancer care**	
Yes	24 (11.1)
No	183 (84.7)
*Prefer not to say*	9 (4.2)
**Have received support from primary cancer hospital for mental health during COVID-19**	
Yes	10 (4.6)
No & Did not need support	199 (92.2)
*Prefer not to say*	7 (3.3)
**Wanted more support for mental health during COVID-19**	
Yes	15 (6.9)
No	192 (88.9)
*Prefer not to say*	9 (4.2)
**Personal coping strategies**	
Yes ^a^	
Focussing on positives	115 (53.2)
Using humour	84 (38.9)
Change in physical activity (e.g., exercise)	73 (33.8)
Avoiding thinking about it	66 (30.6)
Planning time	50 (23.1)
Distracting self	43 (19.9)
Changes in diet (e.g., types of food, amount)	28 (13.0)
Using religious or spiritual practice(s)	26 (12.0)
Talking to medical professions	24 (11.1)
Using meditation, mindfulness, or other relaxation techniques	21 (9.7)
Changing substance intake (e.g., smoking, alcohol, other drugs)	11 (5.1)
Other	2 (0.9)
None of the above	38 (17.6)
*Did not answer*	1 (0.5)

^a^ more than one option to question possible.

**Table 3 cancers-15-01226-t003:** Univariate and multivariate analyses of factors associated with anxiety (GAD-7 ≥ 5), depression (PHQ-9 ≥ 10), and poor well-being (WHO-5 < 50).

Variable	Univariate Analysis			Multivariate Analysis		
	Odds Ratio	95% CI	*p* Value	Odds Ratio	95% CI	*p* Value
**Factors Associated with Anxiety (GAD-7 ≥ 5)**						
Age (years)	0.963	0.934–0.993	0.016			
Ethnicity (White vs. all/others)	0.35	0.12–1.01	0.062			
Concerned regarding cancer treatment	2.88	1.41–5.88	0.004			
Concerned might get COVID-19	1.33	0.99–1.78	0.058			
**Effect on mental health**	**2.61**	**1.82–3.73**	**<0.001**	**2.26**	**1.32–3.88**	**0.003**
**Support from Christie**	**5.38**	**1.45–19.97**	**0.006**	**9.77**	**1.39–68.93**	**0.022**
Mental health affected care	8.59	3.35–22.03	<0.001			
Wanted more support	7.30	2.31–23.11	0.001			
Past history: Nervous/Anxious	4.03	1.67–9.73	0.002			
Past history: Depression	4.94	1.73–14.12	0.001			
**Past history: None of the above**	**0.28**	**0.13–0.60**	**0.001**	**0.17**	**0.06–0.50**	**0.001**
Coping: Distracting self	2.13	1.00–4.52	0.047			
Coping: Avoiding thinking about it	2.5	1.27–4.90	0.007			
Coping: Changes in diet	2.63	1.08–6.39	0.028			
Coping: None of the above	0.32	0.11–0.96	0.035			
Self-reported perception of current status of cancer Progressive disease/Stable disease	2.05	0.85–4.93	0.110			
**Self-reported perception of current status of cancer** **Unknown/Stable disease**	**2.42**	**1.09–5.38**	**0.030**	**5.16**	**1.56–17.10**	**0.007**
Support: Cancer team	0.69	0.51–0.95	0.022			
Support: Community services	0.80	0.64–1.00	0.054			
Support: Government	0.79	0.60–1.04	0.091			
Lack of contact with clinical team	3.19	0.88–11.56	0.064			
**Factors Associated with Depression (PHQ-9 ≥ 10)**						
**Ethnicity (White vs. other)**	**0.24**	**0.07–0.77**	**0.010**	**0.12**	**0.02–0.90**	**0.039**
Effect on mental health	2.37	1.58–3.54	<0.001			
Concerned that COVID-19 had/will have a negative impact on their cancer treatment (Q10)	2.24	0.92–5.45	0.075			
**Mental health affected care**	**9.55**	**3.62–25.22**	**<0.001**	**6.18**	**1.15–33.03**	**0.033**
Wanted more support	25.69	7.64–86.40	<0.001			
Past history: Nervous/Anxious	2.93	1.03–8.32	0.037			
Past history: Depression	5.00	1.65–15.13	0.002			
Past history: None of the above	0.36	0.14–0.89	0.022			
Self-reported perception of current status of cancer Progressive disease/Stable disease	7.30	2.37–22.53	<0.001			
Self-reported perception of current status of cancer Unknown/Stable disease	4.15	1.35–12.75	0.008			
Comorbidities (yes/no)	2.69	1.09–6.65	0.027			
Coping: Changes in diet	2.91	1.09–7.78	0.028			
Coping: Talking to medical professionals	3.94	1.43–10.84	0.005			
**Coping: Distracting self**	**3.16**	**1.30–7.69**	**0.008**	**5.57**	**1.12–27.82**	**0.036**
**Coping: Focusing on positives**	**0.29**	**0.12–0.73**	**0.006**	**0.04**	**0.01–0.25**	**<0.001**
Coping: Avoiding thinking about it	2.07	0.88–4.87	0.091			
Coping: None of the above	0.17	0.02–1.30	0.088			
Support: Government	0.71	0.49–1.03	0.072			
Where to get help with dealing with side effects	3.83	1.19–12.33	0.033			
**Factors Associated with Poor Well-Being (WHO-5 < 50)**						
Concerned might get COVID-19	1.36	1.05–1.77	0.021			
Concerned that COVID-19 had/will have a negative impact on their cancer treatment	1.77	0.92–3.43	0.086			
**Effect on mental health**	**1.87**	**1.38–2.56**	**<0.001**	**2.44**	**1.58–3.76**	**<0.001**
Mental health affected care	4.08	1.66–10.01	0.002			
Wanted more support	4.78	1.53–14.92	0.007			
**Coping: Using humour**	**0.34**	**0.18–0.66**	**0.001**	**0.24**	**0.09–0.61**	**0.003**
Coping: Focusing on positives	0.40	0.22–0.72	0.003			
Coping: Planning time	0.50	0.23–1.06	0.068			
**Coping: Changing substance intake (e.g., smoking, alcohol, other drugs)**	**2.84**	**0.84–9.66**	**0.083**	**7.69**	**1.25–47.19**	**0.028**
Lack of contact with clinical team	3.04	0.93–9.99	0.056			
Support: Cancer team	0.65	0.48–0.88	0.005			
Support: Specialist nurse	0.63	0.51–0.80	<0.001			
Support: GP	0.79	0.64–0.97	0.022			
Support: Government	0.73	0.57–0.94	0.017			
**Support: Friends/family**	**0.64**	**0.44–0.92**	**0.016**	**0.52**	**0.33–0.83**	**0.006**

Variables with *p* < 0.10 on univariate analyses were included in the final model respectively. Variables with *p* < 0.05 on multivariate analyses are highlighted in bold. Abbreviations: GAD-7, Generalized Anxiety Disorder scale; PHQ-9, Patient Health Questionnaire-9; WHO-5, World Health Organization Well-being Index.

## Data Availability

All data presented in this study is contained within the article and Appendix A. Any further data, which may be omitted due to privacy or ethical restrictions, may be made available upon reasonable request.

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
