# Peer review of "The Mental Health Burden of Patients with Colorectal Cancer Receiving Care during the COVID-19 Pandemic: Results of the PICO-SM Study"

_cancers, 2023, doi:10.3390/cancers15041226_

Round 1

Reviewer 1 Report

Interesting study aiming to evaluate the impact of the COVID-19 pandemic on the mental health of patients with colorectal cancer. 

The authors describe as a limitation the fact that the study was conducted at a tertiary cancer centre and link this to the availability of support services. But could it also be that the patient population is different at a tertiary cancer centre (more advanced stage, more co-morbidity) and that this has an effect on the mental health of patients? 

The study focusses on colorectal cancer To what extent are the results generalizable to other malignancies? Could the authors make a comparison with results of studies of other tumour types? 

Minor:

- In table 1 there is a “a” (more than one option to question is possible) stated by Self-reported perception of current status of cancer. I don’t think that is correct. The numbers of the answers add up to the total study population. 

- In row 179 and 182 is referred to Table 2, but this information is not in Table 2. 

Author Response

Author Response to Reviewer 1: Point-by-Point

Interesting study aiming to evaluate the impact of the COVID-19 pandemic on the mental health of patients with colorectal cancer. 

We thank Reviewer 1 for their interest in the PICO-SM study, and support for consideration of publication in Cancers

The authors describe as a limitation the fact that the study was conducted at a tertiary cancer centre and link this to the availability of support services. But could it also be that the patient population is different at a tertiary cancer centre (more advanced stage, more co-morbidity) and that this has an effect on the mental health of patients? 

We agree that this is a complex subject matter, and that multiple factors may play a role in the overall impact of the COVID-19 pandemic on patients. As such, we believe that we have highlighted this when discussing the intrinsic limitations of the PICO-SM study design or population of interest (lines 382 to 404).

The study focusses on colorectal cancer To what extent are the results generalizable to other malignancies? Could the authors make a comparison with results of studies of other tumour types? 

We agree with Reviewer 1 that indeed it would be very interesting to compare with other tumour types. With regards to generalization to other tumour types, however, we have made a cautious effort to not infer too much in this direction, so as not to over-conclude based on our single-centre study findings. We have made a broad comparison in this line of thought in our Discussion section, in lines 323 to 326, referring to Ayubi et al. 2021's (ref. no. 19) systematic review and meta-analysis of the characteristics of depression and anxiety among patients with a broad spectrum of cancer types during the COVID-19 pandemic. Similarly, parallelisms have also been drawn for general well-being (lines 334 to 336) and PTSD (lines 344 to 352). We believe that this allows us to contextualise the findings of our study without over-interpreting our data.

Minor Points

In table 1 there is a “a” (more than one option to question is possible) stated by Self-reported perception of current status of cancer. I don’t think that is correct. The numbers of the answers add up to the total study population. 

We thank Reviewer 1 for their detailed review of our data. We have reviewed our source data files, and agree that this is indeed a mistake and have deleted "a" after the heading 'Self-reported perception of current status of cancer' in Table 1.

In row 179 and 182 is referred to Table 2, but this information is not in Table 2. 

We thank Reviewer 1 for their vigilance and apologise for this error. Indeed, the data described in this paragraph (rows 186 to 194) are from Supplementary Table S2. This has now been amended in the tracked revised version.

We would like to thank Reviewer 1 once again for their time and careful peer review of my manuscript, and we believe that we have now addressed their comments. We would be more than happy to discuss further if they have any further questions or feedback.

Reviewer 2 Report

The authors have published their single-center, prospective survey study conducted at a comprehensive cancer centre in Manchester, UK, which demonstrates impressive psychologic and psychiatric burden among patients with colorectal cancer.  The study is well designed in that it is prospective, and compares respondents' answers to well validated tools such as the GAD-7 and PHQ-9. Kudos to their efforts.

Major Comments:

1) "Consent was implied upon return of the 105 completed survey." lines 105-106-- can you clarify how the local IRB affirmed this study method rather than signing a traditional consent form?

2) The sample size is overwhelming "White/White British." Can the authors comment on the overall demographics of the Manchester cancer center's colorectal oncology clinic, and if there was a sample bias in the successfully screened and invited participants? The discussion on lines 326-328 only briefly mentions this.

3) The baseline psychiatric comorbidity of the cohort is impressive: over one-third are listed to have experienced previous anxiety, depression, or panic attacks. Can the authors further comment on this affecting their findings? This is only briefly mentioned, for example, in lines 223-225, and 333-336.

4) Lines 340-343 read confusingly. Are the authors suggesting that cancer centers should not hope for social connections to bolster patients with cancers during future pandemic periods?

Minor Suggestions:

1) Instead of representing the mean age as 65 +/- 10 years in the Abstract and Result sections, it may be easier for the reader to see the mean age alone with the statistical deviations presented in Table 1 instead rather than its currently location in Supplementary Table 2.

2) Please re-phrase the colloquial "getting COVID-19" (e.g. line 176) with a more formal expression related to acquisition and/or transmission.

3) There is a typo and lack of data on line 323: "insert %"

4) There are multiple typos in Supplementary Table 3: the verb "affect" rather than the noun "effect."

Author Response

Author Response to Reviewer 2: Point-by-Point

The authors have published their single-center, prospective survey study conducted at a comprehensive cancer centre in Manchester, UK, which demonstrates impressive psychologic and psychiatric burden among patients with colorectal cancer.  The study is well designed in that it is prospective, and compares respondents' answers to well validated tools such as the GAD-7 and PHQ-9. Kudos to their efforts.

We would like to thank Reviewer 2 for the kind comments and support for the PICO-SM study. This is a collaborative multidisciplinary team effort, including incorporating feedback from patient representatives in the study design.

Major Comments:

1) "Consent was implied upon return of the 105 completed survey." lines 105-106-- can you clarify how the local IRB affirmed this study method rather than signing a traditional consent form?

We thank Reviewer 2 for raising this point of clarification. We agree that the phrasing of this consent process sounds casual, and we have therefore now rephrased this paragraph to better reflect our professional practice (lines 105-111). In addition, we have also submitted blank/unfilled copies of the Patient Information Sheets [PICo SM study V2.0 08/Jun/2021 (IRAS ID - 292413)] and Survey Questionnaire [Version 2.0 dated 08/Jun/2021] (confidential, unpublished), to the Cancers Editorial Team as evidence. The traditional consent form has not been used to allow scope in the study for the participant to remain entirely anonymous, if they choose to do so, and also adapted during this concurrent ‘Lockdown’ period in the UK. The details of the IRB approval are outlined in lines 450 to 453. Please kindly request the Cancers Editorial Team to allow access to these documents for your perusal if required.

2) The sample size is overwhelming "White/White British." Can the authors comment on the overall demographics of the Manchester cancer center's colorectal oncology clinic, and if there was a sample bias in the successfully screened and invited participants? The discussion on lines 326-328 only briefly mentions this.

We thank Reviewer 2 for highlighting this critical point. Reviewer 2 is right that the sampled participants have predominant representation of "White/White British". Prompted by this, we have now systematically reviewed the demographic details of the general population in the geographical region our Colorectal Oncology Service cover and overview of our new patient referrals in 2021.

In the most recent government census (Census 2021, Office for National Statistics; available from: https://www.nomisweb.co.uk/sources/census_2021/report?compare=E12000002#section_8 ), the total population of the North West region of England (catchment population of our cohort of patients with colorectal cancer) is 7.4 million. Of these, 86% (6.3 million) identify their ethnicity as ‘White’. At The Christie NHS Foundation Trust (including our peripheral clinic sites), we received 1,038 new patient referrals to our Colorectal Oncology Service in 2021. Approximately 80-90% of our patients are of ‘White/White British’ ethnicity.

Therefore, in fact, the predominance of those from ‘White/White British’ ethnicity amongst our participants is reflective of the population make-up of the geographical region our Colorectal Oncology Service cover. Nevertheless, this does not ignore the fact that ethnic minorities remain underrepresented in our study. During the conduct of this study, we simply invited all consecutive patients, without any other specific exclusion criteria apart from those lacking capacity (lines 78 to 81), during the pre-determined enrolment period (7th to 28th April 2021). This brief period of 3 weeks might have also stochastically resulted in this inadvertent sampling bias.

To reflect the above points, we have included further sentences in the limitation section (lines 389 to 397):

“As discussed previously, we have also noted an underrepresentation of ethnic minorities engaged in our survey. This is however representative of the demographic composition of the North West of England general population (catchment area of our Colorectal Oncology Service), of whom 86% are of ‘White’ ethnicity (Census 2021, Office for National Statistics). In addition, the PICO-SM study was only open for recruitment for three weeks, and this might have further compounded on the unintentional stochastic sampling bias. Therefore, the potential mental health issues or coping strategies used by this important subgroup of patients remain poorly understood by our study.”

3) The baseline psychiatric comorbidity of the cohort is impressive: over one-third are listed to have experienced previous anxiety, depression, or panic attacks. Can the authors further comment on this affecting their findings? This is only briefly mentioned, for example, in lines 223-225, and 333-336.

We thank Reviewer 2 for highlighting this point of discussion. In fact, only one in five (19.4%, n = 42/216) [rather than over one-third] self-reported having an underlying diagnosis of mental health condition (Table 1, and outlined in Results lines 143-152 and Discussion lines 360 to 366). In this question, participants were allowed to select more than one options i.e. some participants had reported more than one mental health conditions. Hence a 'Yes' for this count = number of individual participants with at least one mental health condition.

However, we also acknowledge that the formal diagnosis of these self-reported mental health conditions cannot be confirmed by the modality in which this anonymous questionnaire had been administered (lines 363 to 366). 

4) Lines 340-343 read confusingly. Are the authors suggesting that cancer centers should not hope for social connections to bolster patients with cancers during future pandemic periods?

We thank Reviewer 2 for requesting further clarification of these lines. After re-reading this, we can understand how this may be confusing. We have therefore rephrased this to crystalise the intended meaning (lines 367 to 372):

"Strong social support has been generally linked with increased psychological resilience after a global disaster (25). In our cohort, all but few participants felt moderately to extremely well-supported by their friends and family, which might account for why the majority did not feel they needed more support. However, healthcare authorities should not rest on their laurels and continue to maintain high levels of planning and preparedness to meet patients’ demands during any potential crisis period in the future."

Minor Suggestions:

1) Instead of representing the mean age as 65 +/- 10 years in the Abstract and Result sections, it may be easier for the reader to see the mean age alone with the statistical deviations presented in Table 1 instead rather than its currently location in Supplementary Table 2.

We thank Reviewer 2 for this suggestion, and have removed the SD from the Abstract (line 34) and Result (line 142). We have checked that the SD has indeed been included in Table 1 ('65.3±10.2 years', n = 216 - primary cohort) and Supplementary Supplementary S2 ('65.0 ± 9.1 years', n = 95 - tracked cohort).

2) Please re-phrase the colloquial "getting COVID-19" (e.g. line 176) with a more formal expression related to acquisition and/or transmission.

Apologies for this oversight. Formalise language now used: e.g. in lines 187 and 275, 'getting' - replaced by 'contracting'. We have now gone through the manuscript carefully and have made changes to other potential colloquialism used, but would be happy to make further revisions if necessary.

3) There is a typo and lack of data on line 323: "insert %"

Apologies for accidental omission. This pertains to line 351, and refers to the data reported in Joly et al. 2021 study (ref. no. 16) - this missing data has now been inserted: '14%'.

4) There are multiple typos in Supplementary Table 3: the verb "affect" rather than the noun "effect."

We apologise to Reviewer 2 for these honest mistakes/typos, and have further proof-read all the tables in the revised manuscript. Grammatical errors and typos have now been corrected in the revised Supplementary Table S3, and similar edits have been made in main Table 3.

We would like to thank Reviewer 2 for their time and detailed review of my manuscript, and we believe that we have now addressed their comments. We would be more than happy to discuss further if they have any further questions or feedback.
